

# Subglacial hydrology modulates basal sliding response of the Antarctic ice sheet to climate forcing

Elise Kazmierczak[1], Sainan Sun[1,2], Violaine Coulon[1], and Frank Pattyn[1]

[1]Laboratoire de Glaciologie, Université libre de Bruxelles (ULB), Brussels, Belgium
[2]Department of Geography and Environmental Sciences, Northumbria University, Newcastle upon Tyne, UK

**Correspondence:** E. Kazmierczak (Elise.Kazmierczak@ulb.ac.be)

**Abstract.** Major uncertainties in the response of ice sheets to environmental forcing are due to subglacial processes. These processes pertain to the type of sliding or friction law as well as the spatial and temporal evolution of the effective pressure at the base of ice sheets. We evaluate the classical Weertman/Budd sliding law for different power exponents (viscous to near plastic) and for different representations of effective pressure at the base of the ice sheet, commonly used for hard and soft beds. The sensitivity of above slip laws is evaluated for the Antarctic ice sheet in two types of experiments, i.e., (i) the ABUMIP experiments in which ice shelves are instantaneously removed, leading to rapid grounding line retreat and ice sheet collapse, and (ii) the ISMIP6 experiments with realistic ocean and atmosphere forcings for different RCP scenarios. Results confirm earlier work that the power in the sliding law is the most determining factor in the sensitivity of the ice sheet, where a higher power in the sliding law leads to increased mass loss for a given forcing. Here we show that spatial and temporal changes in water pressure or water flux at the base modulates basal sliding for a given power. In particular, subglacial models depending on subglacial water pressure decrease effective pressure significantly near the grounding line, leading to an increased sensitivity for a given power in the sliding law.

## 1 Introduction

The Antarctic ice sheet (AIS) is the biggest fresh water storage on Earth (Fretwell et al., 2013; Rignot et al., 2019), and has been losing mass at an accelerating pace due to an increased ice discharge of the West-Antarctic ice sheet (WAIS; Rignot et al., 2019; Shepherd et al., 2019). Future mass loss of the ice sheet represents the largest source of uncertainty with respect to future sea level rise (Kopp et al., 2014, 2017; Frederikse et al., 2020). Major uncertainties pertain to the interaction of the Antarctic ice sheet with the ocean (Alley et al., 2015; Scambos et al., 2017; de Boer et al., 2017; Asay-Davis et al., 2017), but this interaction is modulated by internal ice sheet dynamics that govern the rate and magnitude of ice sheet mass change in response to changes at the marine boundary. Internal ice sheet dynamics are dominated by the basal conditions of the ice sheet and more precisely the amount and type of basal sliding that takes place (Pattyn, 1996; Ritz et al., 2015; Pattyn, 2017; Bulthuis et al., 2019; Joughin et al., 2019; Sun et al., 2020). Especially, the contrast between viscous (linear) and plastic (Coulomb) sliding law makes the latter far more responsive to changes at the marine boundary. In addition to the sliding type, physical basal conditions, such as basal temperature conditions, bed properties (hard or soft), subglacial water flow and drainage, or





till properties and mechanics, add to the sensitivity of ice sheet flow (Clarke, 2005; Cuffey and Paterson, 2010). While the presence of subglacial water leads to lubrication between the ice and the bed, high subglacial water pressure is implicated in persistent and episodic fast ice flow processes (Bindschadler, 1983; Tulaczyk et al., 2000). However, few have attempted to establish physical laws coupling subglacial hydrology and basal sliding on a large scale, such as ice sheets (Flowers, 2015).

Nevertheless, our understanding of subglacial processes has greatly improved, either from a theoretical viewpoint (Schoof
et al., 2012; Hewitt et al., 2012), or through the development of high-resolution subglacial hydrological models (Flowers and Clarke, 2002a; Hewitt, 2011; Werder et al., 2013; Fleurian et al., 2014; Hoffman and Price, 2014; Bueler and van Pelt, 2015; Brinkerhoff et al., 2016; Beyer et al., 2018; Gagliardini and Werder, 2018; Sommers et al., 2018), eventually leading to the first subglacial model intercomparison SHMIP (de Fleurian et al., 2018). Despite these efforts, applying subglacial models on the scale of ice sheets remains challenging due to the large computational demand and remains therefore mostly limited to
individual basins on short time scales (Bougamont et al., 2014; Dow et al., 2016). In addition, subglacial processes in large-scale ice sheet models are limited by spatial and temporal scale and therefore rely on large-scale representations of such processes (Le Brocq et al., 2009; Flowers, 2015). Moreover, the lack of direct observations or precise measurements of the subglacial system makes their understanding particularly difficult. Many models deal with the hydrological system in an implicit way through data assimilation of observed satellite velocities to retrieve spatially varying basal friction coefficients. While this
leads to improved initial conditions of the ice sheet, it does not allow for temporal changes in the hydrological system and and to gauge their effect on the sliding behaviour of the ice sheet.

Here, we consider four different ways, commonly found in the literature and generally used in large-scale ice sheet models, to link the hydrology to sliding at the base of ice sheets (Alley, 1989; Bueler and Brown, 2009; Winkelmann et al., 2011; Bueler and van Pelt, 2015; Goeller et al., 2013). They encompass the presence of a water film (Weertman, 1957; Le Brocq
et al., 2009) across a rigid bed made of non-deforming rock (a 'hard' bed) and the deformation of a saturated till layer (a 'soft' bed) (Bueler and van Pelt, 2015; Muto et al., 2019). The effective pressure is then used as a boundary condition to the common Weertman/Budd (Weertman, 1957; Budd and Jenssen, 1987) sliding law for different exponents of the power law. Using the f.ETISh ice sheet model of intermediate complexity (Pattyn, 2017; Pelletier et al., 2022), centennial-scale simulations are performed for the ISMIP6 (Seroussi et al., 2020) and ABUMIP (Sun et al., 2020) setups. Results are subsequently analyzed
for the whole ice sheet as well as for separate drainage basins in view of their different characteristics (marine basins, hard/soft bed, land-based ice sheet, etc.).

## 2  Model description

We employed f.ETISh v.1.6 (Pattyn, 2017; Pelletier et al., 2022), which is a vertically integrated hybrid ice sheet-ice shelf model with full thermomechanical coupling. It combines the shallow-ice and shallow shelf equations similar to Winkelmann
et al. (2011). Input data are the present day ice sheet surface and bed geometry from Bedmachine (Morlighem et al., 2020), surface mass balance and temperature from RACMO2 (Van Wessem et al., 2014), and a prescribed field for the geothermal heat flux (Shapiro and Ritzwoller, 2004). All datasets were resampled at a spatial resolution of 25 km.



**Table 1.** List of constants and parameters used.

| Symbol | Description | Units | Value |
|---|---|---|---|
| $C_c$ | Till compressibility | | 0.12 |
| $C_d$ | Till drainage rate | mm a$^{-1}$ | 1 |
| $d_w^0$ | Maximum limit of subglacial water film thickness | m | $15 \times 10^{-3}$ |
| $\delta$ | Fraction of ice overburden pressure | | 0.02 |
| $e_0$ | Reference value of the sediment void ratio | | 0.69 |
| $\phi_0$ | Water flux limit | m$^2$ a$^{-1}$ | $10^5$ |
| $g$ | Gravitational acceleration | m s$^{-2}$ | 9.81 |
| $m$ | Exponent in basal sliding law | | 1–5 |
| $\mu$ | Water viscosity | Pa s | $1.8 \times 10^{-3}$ |
| $N_0$ | Reference value of the effective basal pressure | Pa | 1000 |
| $p$ | Exponent of the effective pressure | | 0-1 |
| $\rho_i$ | Ice density | kg m$^{-3}$ | 917 |
| $\rho_s$ | Ocean water density | kg m$^{-3}$ | 1027 |
| $\rho_w$ | Water density | kg m$^{-3}$ | 1000 |
| $W_{max}$ | Maximum saturated till layer thickness | m | 2 |

The model uses a power law for basal sliding (Weertman, 1957; Budd et al., 1979; Budd and Jenssen, 1987), i.e.,

$$\tau_b = A_b^{-1/m} N^p |v_b|^{1/m-1} v_b, \tag{1}$$

where $\tau_b$ is the basal shear stress, $v_b$ is the basal sliding velocity, $A_b$ is a spatially-varying basal sliding coefficient, and $N$ is the effective pressure at the base. Values of exponents $m$ and $p$ are generally within the range of 1–3, although higher values for $m$, leading to a more plastic behaviour are also found (Gillet-Chaulet et al., 2016; Brondex et al., 2019). Values for $A_b$ are obtained through a nudging method in which the modelled ice sheet is run in a steady state and slip coefficients $A_b$ adjusted to minimize the difference between modelled and observed ice thickness (Pollard and DeConto, 2012b; Pattyn, 2017). Subglacial hydrology generally enters Eq. (1) through the effective pressure $N$, which makes it part of the nudging scheme mention above. At the ocean boundary, a grounding-line flux condition is employed (Pollard and DeConto, 2012b; Pattyn, 2017) in line with a Weertman sliding law (Schoof, 2007). The flux condition implies that the effective pressure remains non-zero at the grounding line (Tsai et al., 2015), which is the case for all values of $N$ determined in the large-scale experiments. We will discuss this particularity more in depth in the discussion section.





## 2.1 Effective pressure at the ice sheet base

There are different ways of representing the effective pressure at the base of an ice sheet applied in ice sheet models. In theory, the effective pressure $N$ represents the ice overburden pressure $p_o$, i.e., the downward force due to the weight of overlying ice, minus the subglacial water pressure ($p_w$) :

$$N = p_o - p_w = \rho_i gh - p_w \,, \tag{2}$$

where $\rho_i$ is the ice density, $g$ is the gravitational acceleration, and $h$ is the ice thickness. Values for all employed constants are provided in Table 1. In the following sections, we describe in more details how the effective pressure $N$ is determined in space and time, either directly, or by determining the subglacial water pressure $p_w$.

### 2.1.1 Height above buoyancy (HAB)

A simple way to determine subglacial water pressure $p_w$ is to link it directly to the depth of the bed below sea level (Van der Veen, 1987; Tsai et al., 2015; Martin et al., 2011; Winkelmann et al., 2011), so that high water pressure occurs in the deep subglacial basins and near the grounding line (Fig. 1). The subglacial water pressure at the base may then be approximated by:

$$p_w = -P_w \rho_s g \left( b - z_{sl} \right) \,, \tag{3}$$

where $P_w$ is a fixed fraction of the overburden pressure, $\rho_s$ is the density of sea water, $b$ is the bedrock elevation and $z_{sl}$ the local sea level height. Eq. (3) is valid for $b - z_{sl} < 0$, otherwise $p_w = 0$. By definition, $p_w = \rho_i gh$ at the grounding line and underneath floating ice shelves, so that the effective pressure becomes zero (or close to zero when modulated by the value of $P_w$). This means that only marine terminated parts of the ice sheet are impacted by the subglacial water. According to Lüthi et al. (2002), the pore water pressure, i.e., the pressure of the subglacial water mixed with the solid part of the till, represents a fraction slightly smaller than 100% of the ice overburden pressure. Bueler and Brown (2009) consider the pore water pressure locally as at most a fixed fraction ($P_w$ = 95%) of the ice overburden pressure $\rho_i gh$. The fraction varies among different studies, i.e., 96% (Martin et al., 2011; Winkelmann et al., 2011), 97% (Van Pelt and Oerlemans, 2012), and 99% (Gandy et al., 2019). Here, we fixed $P_w = 0.96$. The use of HAB is probably the most common representation of subglacial water pressure in large-scale Antarctic ice sheet models, and the value of $P_w$ prevents the effective pressure to become zero when the flotation criterion is reached (e.g., at the grounding line) to avoid numerical instabilities.

### 2.1.2 Subglacial water depth (SWD)

Subglacial water flow can be introduced following the method of Le Brocq et al. (2009) based on a single element type to describe the morphology of the drainage system, i.e., a Weertman-type water film (Weertman, 1972; Walder, 1982; Weertman





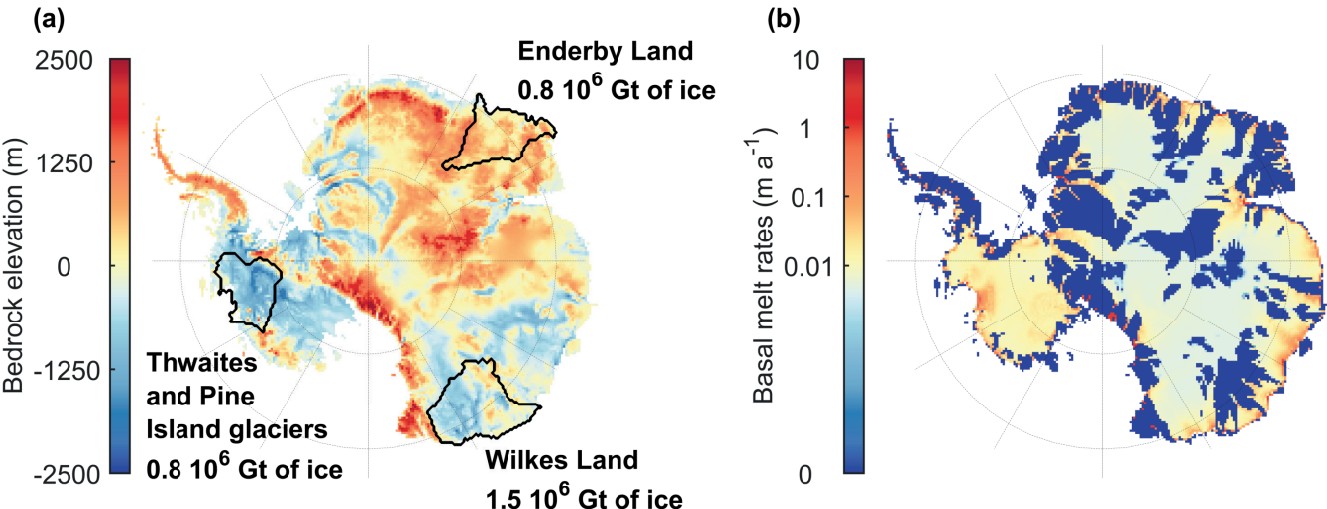

**Figure 1.** Bedrock elevation (m) based on BedMachine data (Morlighem et al., 2020) (a), and calculated basal melt rates (m a$^{-1}$) with the ice sheet model (b). Selected drainage basins shown are discussed in Section 4.2.

and Birchfield, 1982). The model assumes that water flows in a thin film of water of the order of $10^{-3}$ m thickness. The evolution of the water film depth $d_w$ is given by

$$\frac{\partial d_w}{\partial t} = M - \nabla \cdot (\mathbf{u}_w d_w), \tag{4}$$

where $M$ is the basal melt rate (positive for melting) underneath the grounded ice sheet (Fig. 1), and $\mathbf{u}_w$ is the depth-averaged water film velocity, calculated using a theoretical treatment of laminar flow between two parallel plates, driven by differences in the water pressure (Weertman, 1966):

$$\mathbf{u}_w = \frac{d_w^2}{12\mu} \nabla \Phi, \tag{5}$$

where $\mu$ is the viscosity of water and $\Phi$ is the hydraulic potential. The hydraulic potential represents the total mechanical 105      energy per unit volume of water required to lead to spontaneous water flow, and is a function of the elevation potential and the water pressure (Le Brocq et al., 2009), i.e.,

$$\Phi = \rho_w g b + p_w = \rho_w g b + \rho_i g h - N. \tag{6}$$

The water pressure, $p_w$, is a function of the ice overburden pressure and the effective pressure $N$. In a distributed system (opposite of the channelized system (Flowers, 2015)), however, the water pressure will be close to, if not at, the overburden 110      pressure. As a result, a simplification can be made to Eq. (6), assuming $N$ to be zero (Budd and Jenssen, 1987; Alley, 1989).





The assumption that $N$ is zero simplifies the calculation of the hydraulic potential surface by removing the need to calculate the water pressure. With this simplification, the gradient of the potential surface is written as

$$\nabla \Phi = \rho_i g \nabla h_s + (\rho_w - \rho_i) g \nabla b. \tag{7}$$

Taking $\partial d_w / \partial t = 0$ in Eq. (4) (steady-state approach), it is possible to use a flux balance approach to calculate the steady-
state water depth in a similar way to balance velocity calculations (Budd and Warner, 1996; Le Brocq et al., 2006). The balance approach requires the outgoing flux in any given grid cell to be equal to the incoming water flux plus the local melt rate within the cell. The routing direction of the subglacial water is given by the hydraulic potential gradient $\nabla \Phi$. The water depth is then obtained from the outgoing water flux $Q_l$, i.e.,

$$d_w = \left( \frac{12 \mu Q_l}{|\nabla \Phi|} \right)^{\frac{1}{3}}. \tag{8}$$

Subglacial water thickness is then related to subglacial water pressure through

$$p_w = P_w \rho g h \left( \frac{d_w}{d_w^0} \right), \tag{9}$$

where $d_w^0$ = 0.015 m is a limit value to the subglacial water thickness.

### 2.1.3   Sliding related to water flux (SWF)

Alternatively, Goeller et al. (2013) propose to introduce a simple physically plausible correlation of the basal sliding coefficient and the subglacial water flux, i.e.,

$$A_b = A_o \exp\left( \frac{\phi}{\phi_0} \right), \tag{10}$$

where $\phi_0$ is a limit factor on the subglacial water flux (Goeller et al., 2013), and $A_o$ the initial value of $A_b$. A similar approach has been followed by Pattyn et al. (2005). The approach considers that basal sliding increases when the water flux beneath the ice sheet increases which is however more intuitive than physically sound as the water flux generally increases towards the grounding line, hence leading to the highest sliding velocities at the grounding line. Since the subglacial hydrology enters through the basal sliding coefficient, the effective pressure $N$ is considered constant for SWF.





### 2.1.4 Effective pressure in till (TIL)

Bueler and Brown (2009) employ an effective thickness of stored liquid water at the base of the ice column. This layer of thickness $W$ is used to estimate the subglacial water pressure reduced to the pore water pressure according to

$$p_w = P_w \rho_i g h \frac{W}{W_{\max}},\tag{11}$$

where $W_{\max}$ is the maximum saturated till thickness, fixed at 2 m, which has an impact on the till weakening by pressurized water. A fixed fraction of ice overburden equal to one implies that the yield stress becomes zero in the case of full till saturation (Van Pelt and Oerlemans, 2012). An alternative way consists of deriving the effective pressure in the case of a deformable bed composed by a permeable till. The effective pressure, $N$, in Eq. (2) is expressed as a function of the sediment void ratio, $e$, due to the changing water content in the till (van der Wel et al., 2013; Bougamont et al., 2014), i.e.,

$$N = N_0 \times 10^{-(e-e_0)/C_c},\tag{12}$$

where $e_0$ is the void ratio at a reference effective pressure $N_0$ and $C_c$ is the dimensionless coefficient of till compressibility (Tulaczyk et al., 2000). Bueler and van Pelt (2015) propose to employ Eq. (12) in a hydrological model of subglacial water drainage within an active layer of the till, $W$. As the water in till pore spaces is much less mobile than that in the linked-cavity system because of the very low hydraulic conductivity of till, an evolution equation for $W_{\mathrm{til}}$ without horizontal transport can be written (Bueler and van Pelt, 2015)

$$\frac{\partial W}{\partial t} = M - C_t.\tag{13}$$

Here, $C_t$ is a fixed rate that makes the till gradually drain in the absence of water input; we choose $C_t$ to be 1 mm a$^{-1}$ (Pattyn et al., 2005; Bueler and van Pelt, 2015), which is small compared to typical values of subglacial melt. We constrain the layer thickness by $0 \leq W \leq W_{\max}$. The effective pressure $N$ is then written as the following function of $W$ (Bueler and van Pelt, 2015),

$$N = N_0 \left( \frac{\delta p_o}{N_0} \right)^s 10^{\left( \frac{e_0}{C_c} \right)(1-s)},\tag{14}$$

where $s = W/W_{\max}$ and is bounded by $N = \min\{p_o, N\}$, and $\delta p_o$ is the lower bound on $N$, taken as a fraction of the ice overburden pressure.

## 3 Experimental setup

The experimental setup (Fig. 2) starts from a steady-state ice sheet configuration for which a field of basal sliding coefficients $A_b$ are obtained through a nudging method (Pollard and DeConto, 2012a; Pattyn, 2017). To initialize the ice sheet model, we





**Table 2.** Summary of the subglacial models.

| Type | Description | Eq. |
|------|-------------|-----|
| NON | $p = 0$: no effective pressure | |
| HAB | $p_w$ from marine basin depth | (3) |
| SWD | $p_w$ from subglacial water depth | (9) |
| SWF | $A_b$ from subglacial water flux | (10) |
| TIL | $N$ from saturated till | (14) |

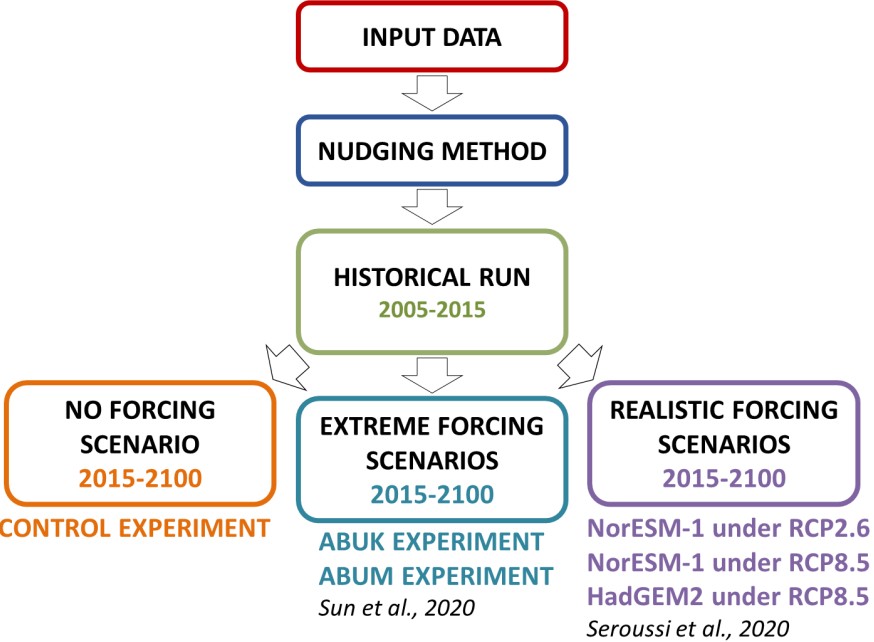

**Figure 2.** Flow chart of the experimental setup.

optimized the basal sliding coefficients $A_b$ to fit the modelled ice sheet geometry to the observed one for four exponents of the sliding law (Eq. 1), ranging from viscous sliding $m = 1$ to near plastic sliding $m = 5$ and $p = 1$ to express the effective

pressure for each of the subglacial models HAB, SWD, SWF and TIL (Table 2). For each of the sliding exponents, a control run (NON) was established for $p = 0$, where the effective pressure is disregarded so that Eq. 1 represents a Weertman sliding law.

A short historical run was performed for a period of 10 years (Fig. 2) forced with conditions of temperature and sub-shelf melt representative of the conditions estimated between 2005 and 2015 (Seroussi et al., 2020), and can thus be considered as a

kind of relaxation run. Starting from the historical run, we performed three types of experiments, i.e., (i) a control experiment



for which climate forcing was not considered (prolongation of the historical run), (ii) the ABUMIP experiments where ice shelves were either removed or were subject to very high sub-shelf melt rates (Sun et al., 2020), and (iii) climate forcing scenarios for different RCPs and for two different climate models (Seroussi et al., 2020). All runs started at the end of the historical run (2015 CE) and ran until the year 2100. The ISMIP6 set enables us to gauge the sensitivity under realistic climate

scenarios, while the ABUMIP set allows us to test the sensitivity of the sliding laws and subglacial models under extreme high forcing and ice sheet collapse. For the historical, control and ISMIP6 runs, sub-shelf melting was determined through the ISMIP6 parameterization based on ocean temperature and salinity for different basins, according to the method explained in Jourdain et al. (2019) and Seroussi et al. (2020).

### 3.1 Historical run

Fig. 3 displays the main characteristics of the different subglacial model approaches for the grounded part of the Antarctic ice sheet after the historical run. The HAB subglacial water pressure, which is related to the bed topography (Fig. 1), reaches the highest values near the grounding line and in the deepest parts of the marine basins of the ice sheet. These high values are prevalent underneath the West Antarctic ice sheet. Water pressure due to subglacial water depths (SWD) is highest were ice flow is concentrated, i.e., in the major ice streams draining the Antarctic ice sheet, and has nonzero values for the areas

where subglacial water is present (where the ice reaches the pressure melting point at the base). A similar pattern is obtained for the SWF model, with stronger concentrations of water fluxes in the downstream parts of the ice streams, especially near the grounding lines. Finally, the TIL model exhibits a quasi constant field of very low effective pressure $N$ for the areas that are at the pressure melting point. The main reason for this particular behaviour is that the till layer –due to generally constant basal temperature conditions– is saturated across these areas, leading to an effective pressure corresponding to saturated till (Fig. 4).

### 185 3.2 ABUMIP experiments

The ABUMIP experiments (Sun et al., 2020) present an idealized case for gauging the sensitivity of sliding laws when extreme forcing is applied. The first experiment (ABUK) consists of removing instantaneously all the ice shelves surrounding the ice sheet at the start of the run, and removing any newly-formed floating ice instantaneously afterwards (so-called 'float-kill'). In other words, at all times, the calving flux is assumed to be larger than the flux across the grounding line to prohibit regrowth

of the shelves. The second experiment (ABUM) is similar, but instead of removing the ice shelves, a constant high value of sub-shelf melt (400 m a$^{-1}$) is applied to all ice shelves.

The ABUMIP experiments previously showed a tendency for increased model sensitivity to the power of the sliding law, leading to more mass loss for a higher power in power-law sliding (Sun et al., 2020). Here, we make a further comparison with different hydrologies for different powers in the Weertman/Budd sliding law.

**Figure 3.** Steady-state subglacial characteristics for the Antarctic ice sheet according to the different subglacial models: (a) HAB subglacial water pressure (Pa); (b) SWD subglacial water pressure (Pa); (c) SWF subglacial water flux ($m^2\ a^{-1}$); (d) TIL effective pressure (Pa).

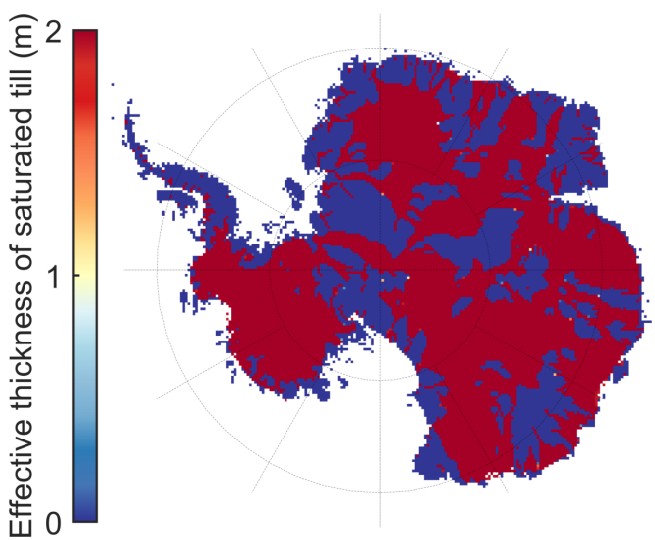

**Figure 4.** Effective thickness of saturated till (maximum value fixed at 2 m) for the TIL experiment.

## 3.3 ISMIP6 experiments

For this set of experiments, we applied three different ISMIP6 forcings in surface mass balance and ocean temperature starting from the historical run (Nowicki et al., 2016; Seroussi et al., 2020). These are NorESM1-M AOGCM for two climate scenarios (RCP2.6 and RCP8.5) and HadGEM2-ES AOGCM for RCP8.5. The latter was chosen for its higher mass loss response of the Antarctic ice sheet (Seroussi et al., 2020). Compared to the ABUMIP experiments, these tests allow for a more realistic future behaviour of the ice sheet, as well as a more in-depth analysis of the subglacial hydrological behaviour for selected basins and bed types.

## 4 Results

### 4.1 ABUMIP experiments

A sudden and sustained loss of ice shelves according to the ABUK experiment (Fig. 5) leads to Antarctic mass loss between 2 and 6.5 m s.l.e. by the end of the century. Mass loss increases with increasing value of $m$ in Eq. (1). The effect of subglacial hydrology modulates this response for each of the values of $m$ and is of the order of 0.5 m around the result of the NON experiment for each value of $m$ (Fig. 5).

Incorporating hydrology in the model generally leads to a higher sensitivity, where SWD and HAB increase the sensitivity of mass loss due to the applied forcing. The subglacial till model (TIL) does not add significant change in response of the ice sheet compared to the absence of basal hydrological coupling (NON). This may be explained by the rather constant value of effective pressure $N$ for TIL across the basal temperate zones of the Antarctic ice sheet (Fig. 3) where mass loss occurs

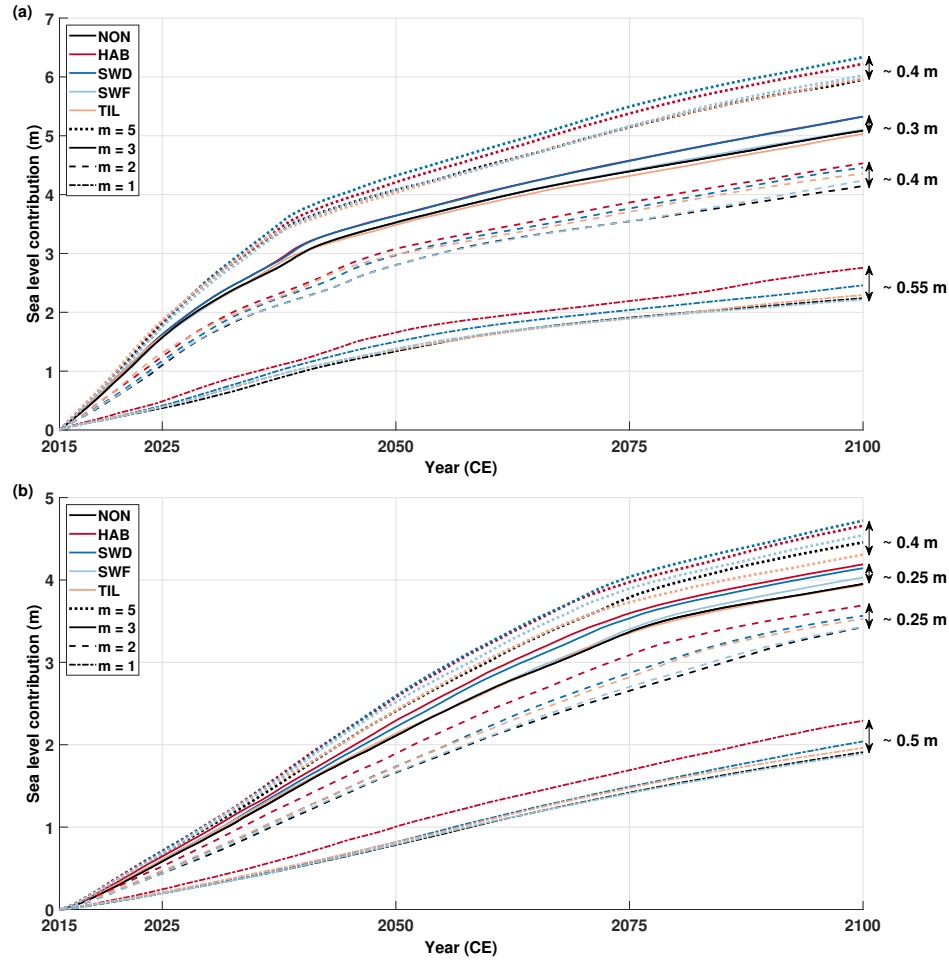

**Figure 5.** Sea level contribution from the Antarctic ice sheet until 2100 following for ABUK (a) and ABUM (b) experiments. Line colors represent the different subglacial models (Table 2) and the different line styles express the exponent value of the basal sliding law.

(Fig. 10). Higher variability in $N$ (or $p_w$), and especially lower values of $N$ within the fast-flowing ice streams, leads to a faster grounding line retreat.

Similar characteristics are shown for the ABUM experiment, where an extreme high sub-shelf melt rate was imposed (Fig. 5).
Here, Antarctic ice mass loss lies slightly lower (between 2 and 5 m by 2100). Again, subglacial hydrological coupling modulates mass loss for different power in the sliding law, similar to what is observed for the ABUK experiment.

## 4.2 ISMIP6 experiments

Results of the ISMIP6 experiments with the NorESM1-M AOGCM forcing are displayed in Figs. 6 and 7 for RCP2.6 and RCP8.5, respectively. To gauge the sensitivity of the different subglacial hydrologies, we subtracted the mean model drift from





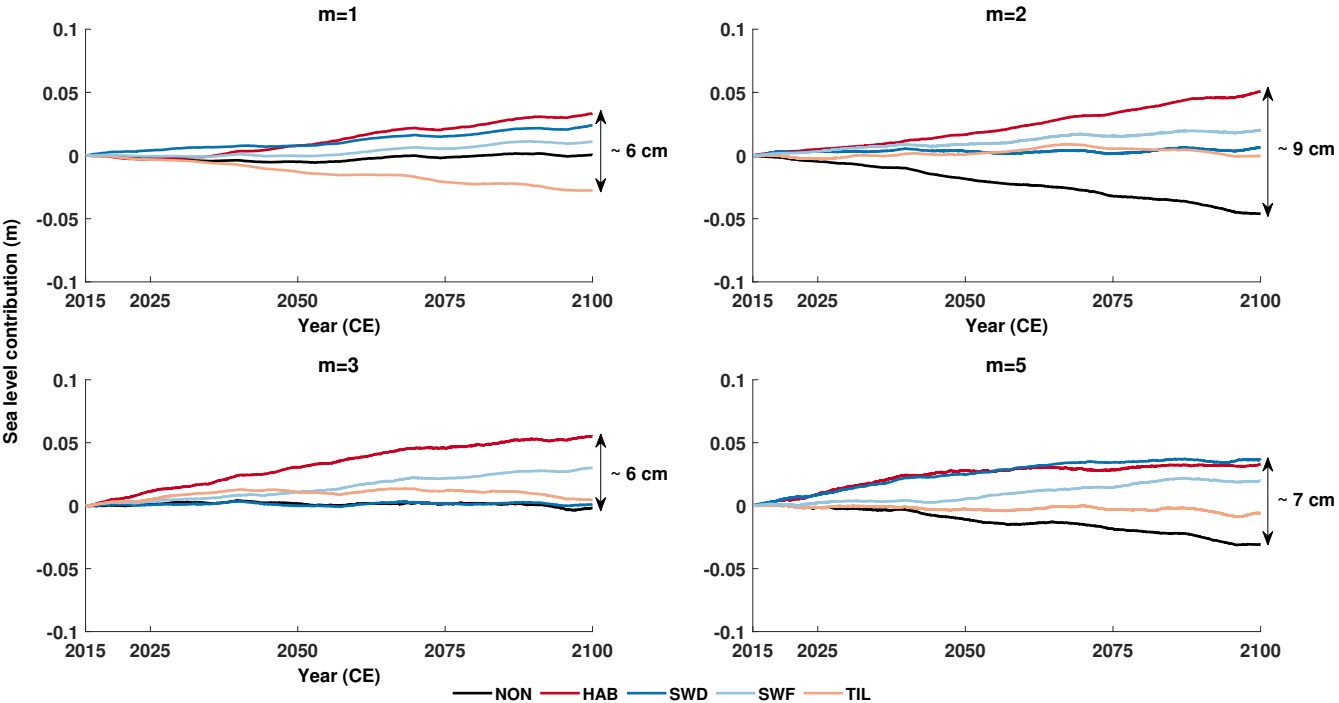

**Figure 6.** Sea level contribution from the Antarctic ice sheet until 2100 following the NorESM-1 in RCP2.6 scenario. Each graph gives the results for one exponent of the basal sliding law and line colors represent the different subglacial models (Table 2).

the control run for each of the experiment series ($m = 1 - 5$) from the time series of the forcing experiments. The drift in the control experiment was observed to increase linearly in time. This way, each of the subglacial approaches has the same model drift correction.

For a low forcing scenario (RCP2.6), model runs with subglacial hydrology lead to either a slight mass gain or close to zero mass change (Fig. 6). Coupling with subglacial hydrology again modulates this signal, characterized by a sensitivity similar to

the ABUMIP experiments, where HAB generally leads to higher mass losses (between 3 and 9 cm by 2100). A similar response is also observed for the high forcing scenario (RCP8.5), making the contribution of the Antarctic ice sheet to sea level change at the end of the century rather independent to forcing (Fig. 6). Similar conclusions were also obtained in a multi-model analysis using the same forcing (Seroussi et al., 2020; Edwards et al., 2021). More striking, however, is that the obvious sensitivity to the power of the sliding law $m$ is absent from both low and high forcing scenarios and that the variability in response is

dominated by hydrological coupling instead of the plasticity of the power law.

Similar to the results from the ISMIP6 model intercomparison, to which the f.ETISh model contributed, NorESM forcing according to RCP8.5 does not significantly differ from the low forcing scenario (Fig. 7) in terms of simulated ice volume above floatation by 2100 (Seroussi et al., 2020). While the values of mass change are slightly larger, the modulation for different subglacial hydrologies is larger as well.





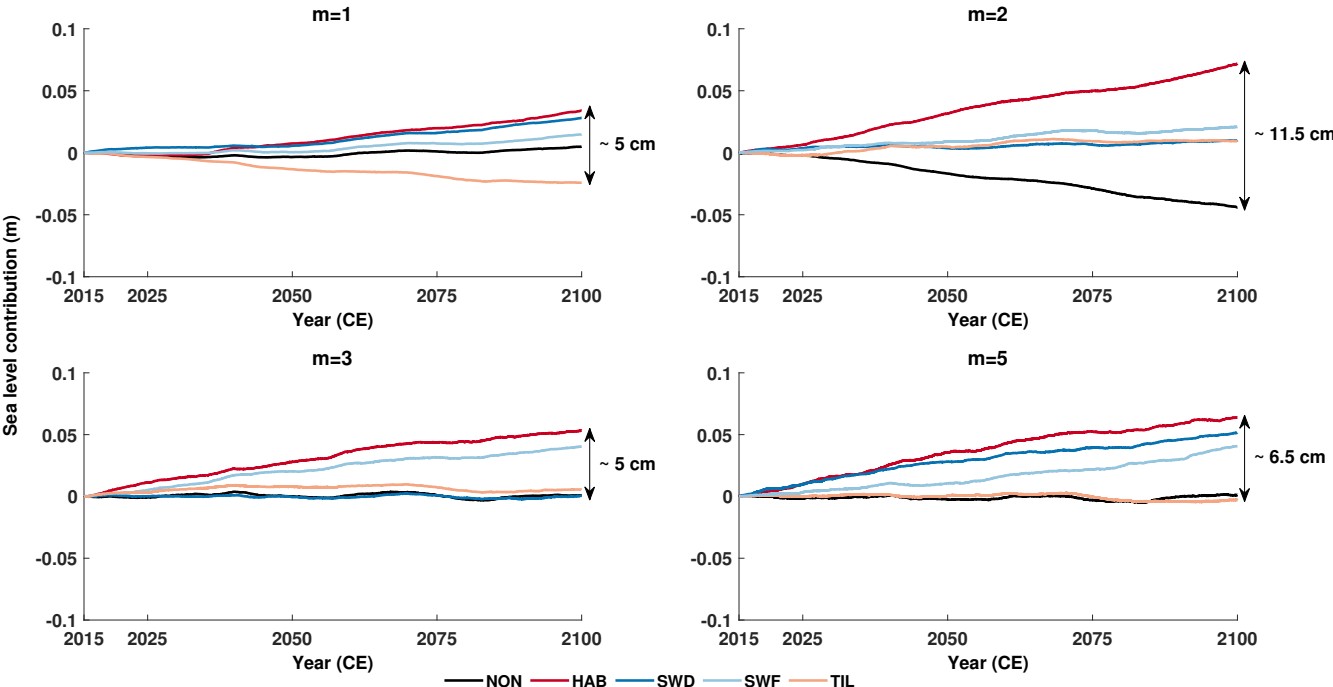

**Figure 7.** Sea level contribution from the Antarctic ice sheet until 2100 following the NorESM-1 in RCP8.5 scenario. Each graph gives the results for one exponent of the basal sliding law and line colors represent the different subglacial models (Table 2).

The HadGEM2 forcing according to RCP8.5 shows a more distinct positive sea level contribution across the different ice sheet models within ISMIP6 (Seroussi et al., 2020). Overall, HadGEM2 forcing leads to higher mass losses, in line with the ISMIP6 results (Fig. 8). In terms of hydrological coupling, HAB is still the most important contributor to ice mass loss, and the variability in sea-level contribution between the different hydrological coupling schemes is also comparable to NorESM.

A more in-depth analysis of the HadGEM2-RCP8.5 results is made for selected drainage basins, each with particular sub-
glacial characteristics, i.e., Thwaites and Pine Island glaciers, Wilkes Land and Enderby Land (Fig. 1). Thwaites and Pine Island are typical marine basins of the West Antarctic ice sheet where most of the current ice mass loss is situated; Wilkes Land is a subglacial basin of the East Antarctic ice sheet with the potential of instability (Mengel and Levermann, 2014), and Enderby Land is part of the East-Antarctic ice sheet where the bed is mostly lying above sea level and where the contact with the ocean is limited (continental ice sheet). All basins contain a more or less comparable ice volume (between 0.8 and 1.5 $10^6$
Gt).

The highest mass loss occurs in the the marine basins of Pine Island and Thwaites Glaciers (Fig. 9), followed by Wilkes basin. Both marine basins have in general the highest response for HAB. Viscous sliding ($m = 1$) has the lowest sensitivity, but higher plasticity of the sliding law does not necessarily result in increasing sensitivity. For instance, Pine Island and Thwaites exhibit the highest sensitivity for $m = 2$, while for Wilkes basin this is for $m = 5$. The complexity in response is
therefore attributable to other factors at play, such as increased accumulation rates and the relatively weak ocean forcing from





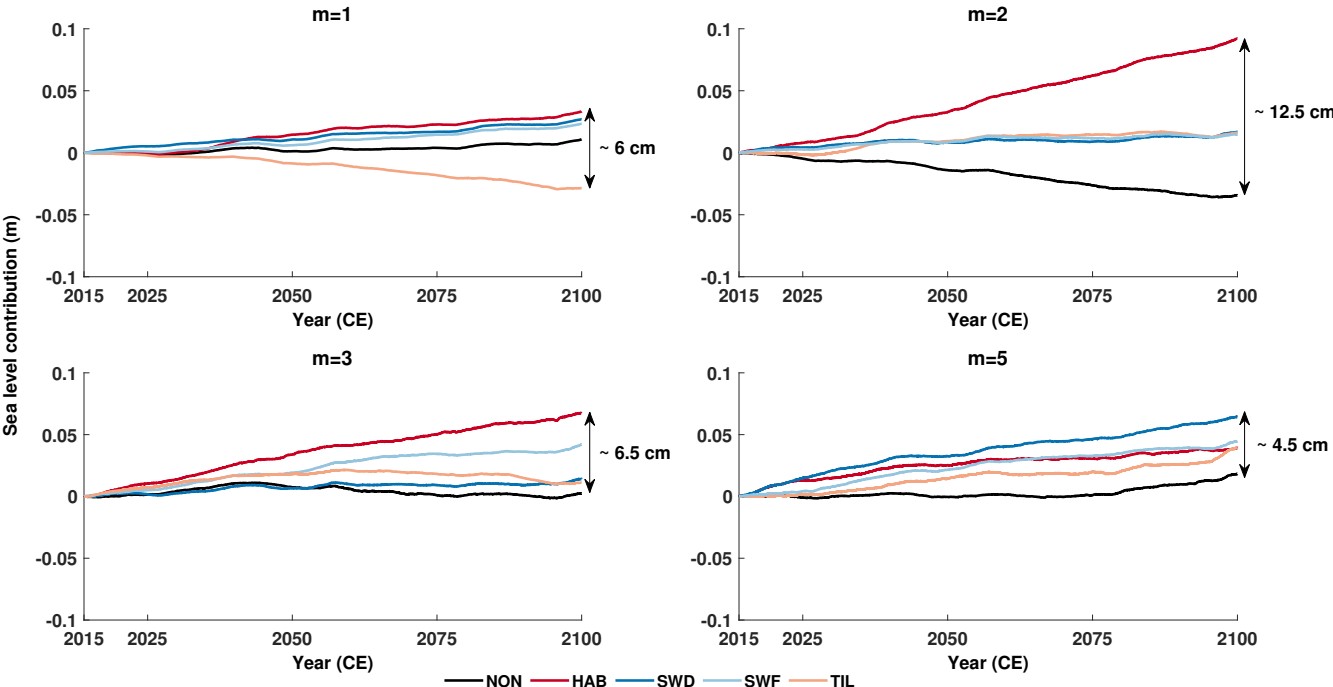

**Figure 8.** Sea level contribution from the Antarctic ice sheet until 2100 following the HadGEM2 in RCP8.5 scenario. Each graph gives the results for one exponent of the basal sliding law and line colors represent the different subglacial models (Table 2).

the HadGEM2 model over this period. The TIL model seems overall more responsive for a higher degree of plasticity of the sliding law. The continental basin (Enderby Land) has only a limited contact with the ocean. Not only are the relative contributions to sea level very minor, any tendency as is seen for the marine basins is also lacking. In summary, on a basin level, subglacial hydrology is more uncertain in altering the response of the ice sheet to forcing from global climate models.

## 5 Discussion

From the above experiments it can be seen that in general, subglacial hydrology increases the sensitivity in response to forcing compared to models where subglacial hydrology is ignored. The uncertainty raised by subglacial hydrology models is comparable to that between difference ice sheet models under RCP scenarios (Seroussi et al., 2020). With non-linear basal sliding laws, an evolving subglacial hydrological system increases the sea level contribution of Antarctica under different RCP scenar-
ios (Fig. 6–8)). This is especially the case for the common HAB model (Van der Veen, 1987; Martin et al., 2011; Winkelmann et al., 2011; Tsai et al., 2015), where the effective pressure at the base of the ice decreases with bed depth and is closest to zero in deep marine basins and near the grounding line (Fig. 3). The SWD model also exhibits a relatively high sensitivity, especially in the ABUK and ABUM experiments. Here, values of effective pressure are generally higher throughout the basins, but low





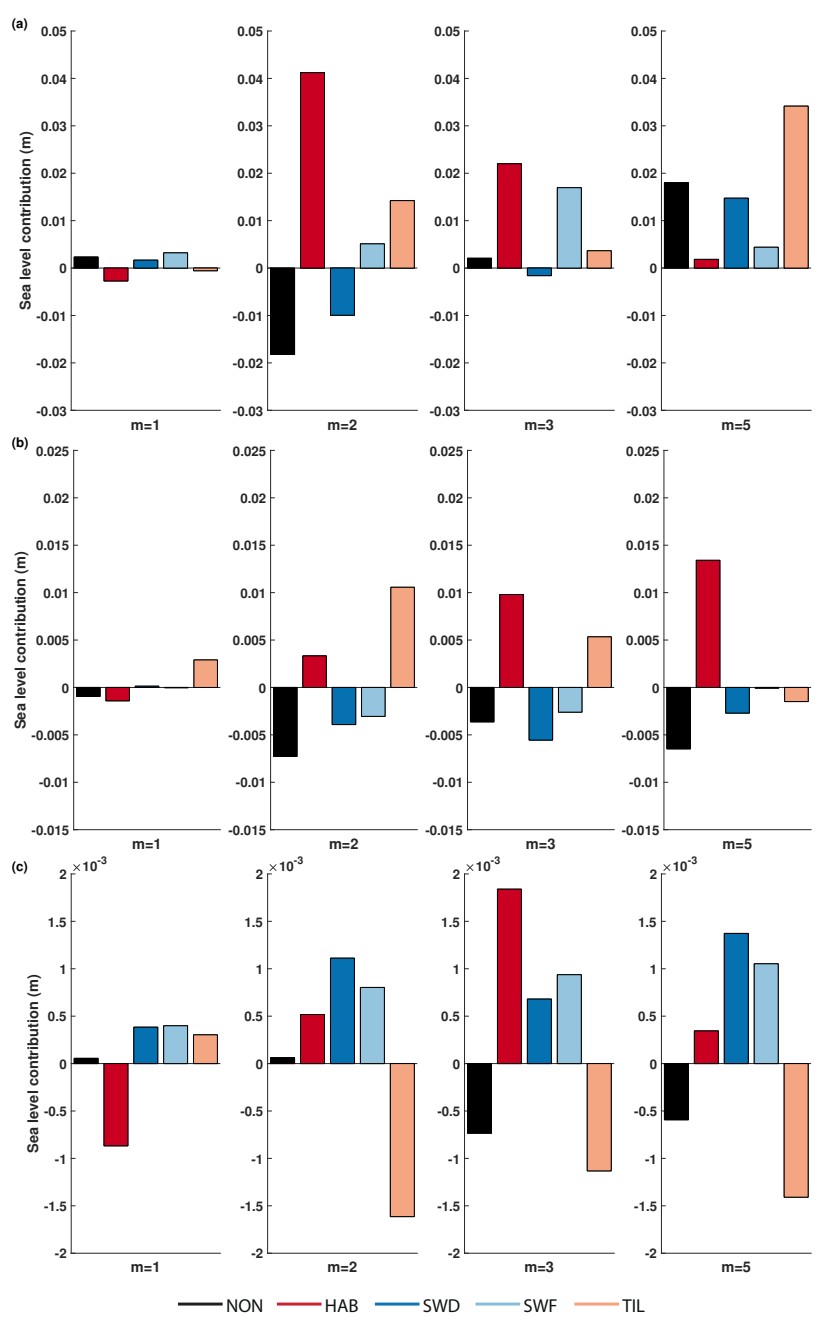

**Figure 9.** Sea level contribution according to HadGEM2-RCP8.5 for different basins, different values of $m$ in the sliding law and subglacial hydrology: Pine Island and Thwaites Glacier (a), Wilkes basin (b), Enderby Land (c). Color bars represent the different subglacial models (Table 2).





values are concentrated in the ice streams and especially close to the grounding line, where it has an impact on the grounding
line sensitivity. All other subglacial models are characterized by higher values of $N$, which leads to a reduced sensitivity.

In principle, the effective pressure becomes zero at the grounding line, as the water pressure equals the ice overburden
pressure (Tsai et al., 2015). For ice shelves, the effective pressure equals zero by definition. However, in large-scale models,
this condition is never really met, as the grounding line is a large grid cell and not exactly a boundary, and limiting factors
are introduced to avoid $N = 0$, which could lead to numerical instabilities. For the Coulomb-limiting case where $N$ becomes
zero at the grounding line, Tsai et al. (2015) found that the grounding-line ice flux depends more strongly on the floating ice
thickness compared to the Weertman sliding case and that the ice sheet may ground stable in shallower water. This implies that
smaller perturbations are required to move the grounding line into regions of retrograde bed slope compared to a power-law
rheology, which makes ice sheets more sensitive to climate perturbations.

In our ice sheet model, a flux condition is employed at the grounding line, corresponding to power-law sliding. This flux
condition is imposed as an internal boundary condition following the implementation by Pollard and DeConto (2012a, b)
based on Schoof (2007). This implementation has been shown to reproduce the migration of the grounding line and correctly
reproduce steady-state grounding line positions for coarse resolution SSA and hybrid SSA/SIA models (Pattyn et al., 2013;
Pattyn and Durand, 2013). Tsai et al. (2015) derived a similar flux condition for the Coulomb friction case where $N = 0$ at
the grounding line. We also implemented this particular case for a linear Coulomb sliding law (Bueler and van Pelt, 2015),
i.e., $\tau_b = CNu$, where $C$ is a friction coefficient related to subglacial till properties. Such friction law is very similar to
the Weertman sliding law with $m = 1$. For this linear case, the grounding line flux is equivalent to $q_g \sim h_g^{n+2}$, compared to
$q_g \sim h_g^{2+n/2}$ for a linear sliding law following the derivation of Schoof (2007). We therefore expect a higher sensitivity of the
grounding line flux to ice thickness for the Coulomb law, which has been shown previously (Bulthuis et al., 2019; Sun et al.,
2020). We obtain similar results for the ABUK and ABUM experiments, i.e., a significant higher mass loss when the Tsai
et al. (2015) grounding line flux is applied. In all cases this leads to the complete collapse of the WAIS within a century and a
consequent grounding line retreat in some EAIS basins, such as the Wilkes basin (Fig. 10). However, the impact of the different
hydrological models follows the same modulation as for the power-law cases, where the HAB model adds to the sensitivity of
the grounding line due to lower effective pressures at the bed.

Only a limited number of subglacial models have been explored in this paper, and the processes on which they are based are
relatively simple. These encompass sliding on hard and soft beds, either through the presence of a water film (SWD, SWF),
elementary till mechanics based on subglacial meltwater saturating the underlying till layer (TIL), or by reduced coupling of
the ice with the bed through lower effective pressure in grounding zones (HAB). Especially the latter is the simplest of all,
but also leads to the highest sensitivity in the different forcing experiments that are presented here. Its validity is limited to
the grounding zone, where it represents the intrusion of sea water reducing the effective pressure in this area. Such process
may well occur tens of kilometers upstream of the grounding line (Golledge et al., 2015; Robel et al., 2022). However, its
applicability further inland across submarine basins can be questioned.

Limits to the models are also due to considering a stable hydrological system. Commonly, a distinction is made between
channelized (efficient) and sheetlike (inefficient or distributed) drainage systems. Here, only the sheetlike system has been





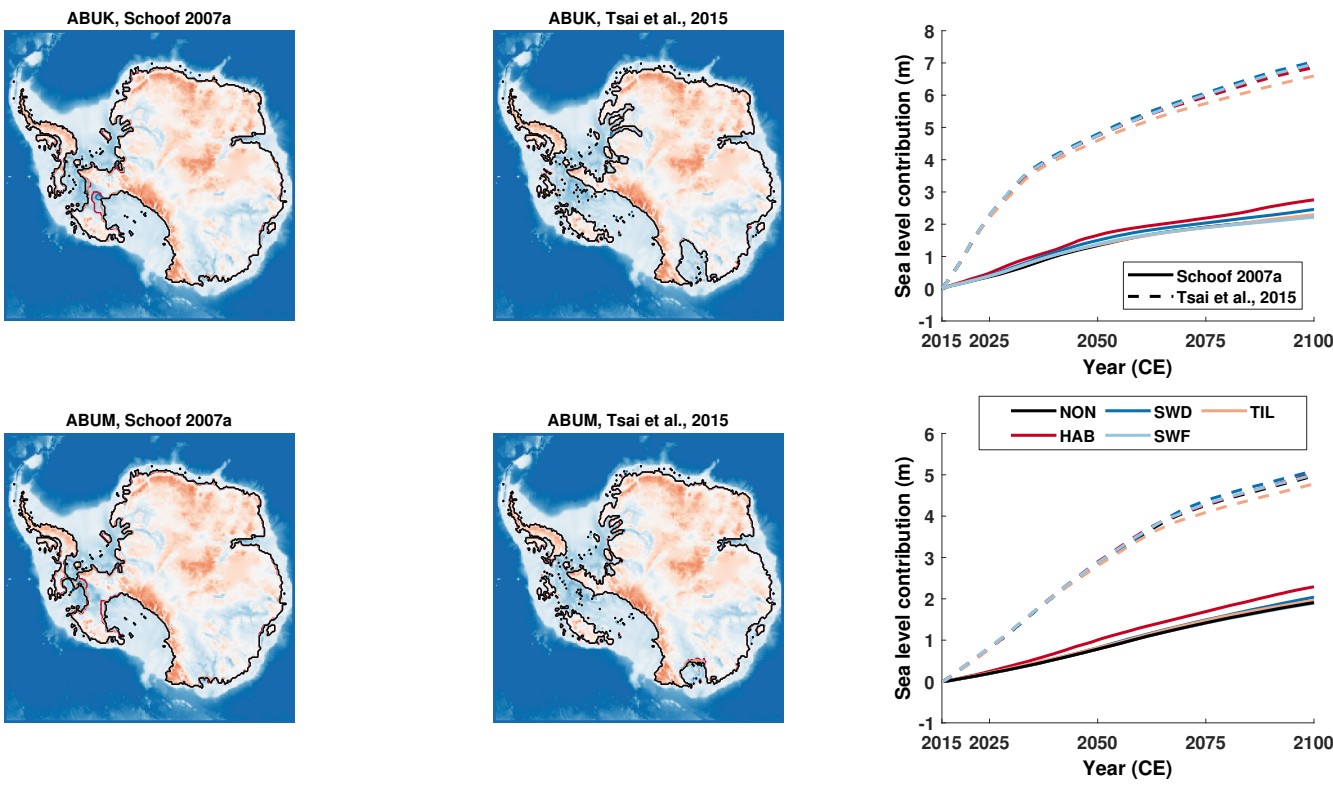

**Figure 10.** Grounding line position at 2100 CE for ABUK and ABUM experiments for both grounding line flux conditions (left panels) and associated mass loss over the same period (right panels). In these experiments the exponent of the sliding law $m$ is equal to 1. Lines color represent the different subglacial models (Table 2), dashed and solid lines show the choice of the parametrization of the flux at the grounding line.

considered, but as a stable system (steady state), thereby intrinsically preventing unstable behaviour, such as the formation of
cavities leading to ice sheet accelerations (Schoof, 2010). This may in part explain the weak sensitivity of SWD and SWF models to the applied forcing.

The evolution of the till-stored water layer thickness (Bueler and Brown, 2009; Bueler and van Pelt, 2015) has also some limitations in the way it is presented here. The evolution of the till layer thickness (Eq. (13)) is limited by a maximum thickness $W_{max}$. For the time scales considered here, the evolution of the subglacial temperate areas is rather constant, leading to a more
or less temporally constant distribution of saturated till thickness in time (Fig. 4). A retreating grounding line will therefore not experience changing basal conditions as the marine basins are already at saturation.

Finally, the SWF model, which is also based on the balanced water layer concept (Le Brocq et al., 2009; Goeller et al., 2013), employs a simple, but physically plausible, correlation of the sliding rate to the subglacial water flux via an exponential function (Eq. (10)). This functional relationship increases basal sliding with increasing basal water flux compared to a baseline
water flux and basal sliding rate. While the approach is probably an oversimplified way of linking both systems, its effect on

the ice sheet behaviour remains rather limited. In most cases it behaves in a very similar way as if hydrological coupling would be omitted.

## 6   Conclusions

We investigated the role of subglacial hydrology and till deformation on the behaviour of the Antarctic ice sheet using a large-
scale ice sheet model. We considered both sliding over a hard bed with the presence of a thin water film and deformation of saturated till at the base of the ice sheet. Both effective pressure models were compared to other common representations of effective pressure in the literature. Our model results confirm that the power in the power-law (Weertman/Budd) sliding law is the most controlling factor determining mass changes of the Antarctic ice sheet, as has been discussed and concluded elsewhere (Brondex et al., 2019; Bulthuis et al., 2019; Sun et al., 2020). Spatial variation of effective pressure $N$ modulates basal sliding
for a given value of the power-law exponent, where reduced effective pressure in the grounding zone generally increases the ice sheet sensitivity. Therefore, subglacial hydrological models that lead to low values of effective pressure in the grounding zone (such as HAB and SWD) increase ice sheet sensitivity to forcing.

In the limit case, where the effective pressure equals zero at the grounding line (Tsai et al., 2015), sensitivity is largely increased and becomes less dependent on the type of sliding law.

*Code and data availability.* The f.ETISh model code can be downloaded from the PARASO source code package (Pelletier et al., 2022, https://doi.org/10.5281/zenodo.5337510). All datasets used in this study are freely accessible through their original reference. We employed BedMachine Antarctica version 2 (https://doi.org/10.5067/E1QL9HFQ7A8M). The ISMIP6 forcing datasets are available from the ISMIP6 website and data portal (https://www.climate-cryosphere.org/wiki/index.php?title=ISMIP6_wiki_page).

*Author contributions.* EK and FP designed the experimental setup and wrote the paper. All experiments were performed by EK with addi-
tional contributions from SS and VC. All authors contributed to the writing of the final manuscript.

*Competing interests.* The authors declare that they have no conflict of interest.

*Acknowledgements.* This project has received funding from the European Union's Horizon 2020 research and innovation programme under grant agreement No 869304, PROTECT contribution number XX. Computational resources have been provided by the Shared ICT Services Centre, Université Libre de Bruxelles.



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
