# Peer review of "Subglacial hydrology modulates basal sliding response of the Antarctic ice sheet to climate forcing"

_The Cryosphere, 2022_

## Author Response (AR1)

**REBUTTAL: Kazmierczak et al. (2022) - TCD**

EDITOR DECISION

Comments to the author:
Dear Authors,

thank you for your replies to the reviewers' comments. While I acknowledge that the referees expressed a positive view of the manuscript, which I largely agree on, it is my feeling that some of the specific comments raised (especially the issue of sensitivity raised by ref. 2, and the connection between model results and conclusions raise by ref 1, as well as my own concern about grounding line flux detailed below) require careful consideration. I would therefore like to invite you to submit a revised version of the paper following a major revision that addresses these points thoroughly. In the hope that you find it helpful and efficient, I am attaching here a set of my own comments, in the hope they can also be addressed in this revision round.

I look forward to receiving your revised manuscript.

With best regards,
Elisa Mantelli

**Dear Elisa,**
**Thank you very much for your comments. Below we answer all of them in bold. We also made the changes in the manuscript and attached a track-change version. We also corrected a few typos that we further came across.**
**kind regards,**
**Elise and co-authors**

Specific comments:

Paragraphs starting at l. 67, and at l. 274:

- I am unclear about what exactly is being done for grounding line flux when effective pressure is accounted for in the friction law (p>0 in eq. 1). The text seems to suggest that that the Schoof (2007) formula is used for all values of p. Is this correct? I am not sure, and if so, I got it only at the end of the paper, so clarification is needed.

- If this is correct though, I have a more substantial concern: the Schoof (2007) formula holds for Weertman friction (that is, the exponent p of effective pressure in eq. 1 is taken to be p = 0). To my understanding this is not the point of the paper, which instead seeks to capture the effects of subglacial hydrology on sliding through (in most simulations) finite p. So then how is the grounding line flux modeled for the case of p>0? As is correctly noted by the authors, a solution for grounding line flux with drainage is offered

by Tsai (2015), which however implies a Coulomb friction law - very different from eq. 1, a so-called Budd law. The Schoof formula does not apply for Budd-type friction, so I am keen on thinking that a simplification must have been introduced. What that is, and on what basis/ in what circumstances it is appropriate needs to be clarified; and I would also want to see an in-depth discussion of implications of this simplification for model results.

**We clarify both remarks at once here. Indeed, the Schoof formula at the grounding line is valid for p=0.  However, technically, it is valid for any value of p as long as N>0. The reason is that we perform an optimization of basal sliding coefficients prior to our runs and imbed N within the sliding prefactor for the use of the Schoof formula. Tests we carried out with a SSA flowline model at high resolution have shown that for nonzero values of N the flux condition of Schoof remains valid for a Budd/Weertman sliding law. In all our experiments, the value of N at the last grounded grid point (even for the very high resolution flowline experiments) remains N>0. As pointed out by Tsai et al. (2015), at the exact position of the grounding line, N becomes zero (as it is by definition zero when ice is floating), but we never reach a value of N=0 at the grounding line. This is explained in lines 64-70 of the original manuscript, and further discussed in the discussion section. However, for the Coulomb friction law used, we apply the Tsai et al (2015) that implies N=0 at the grounding line, which also leads to a higher model sensitivity for the same climatic forcing.**
**We have tried to explain this better and in more detail in the manuscript in two places, i.e., where the flux condition is mentioned in the methods section and in the discussion when a comparison is made with the Tsai flux condition.**

l. 158: maybe a naive question, but why do you choose to fit ice sheet geometry rather than velocities? Can you comment on whether this choice may have any influence on the results at 2100?

**This is definitely not a naive question and good that you pointed this out. Fitting velocities for a given ice geometry is a common approach using inverse methods. However, a drawback is that also an inversion for ice stiffness is needed or that a temperature field is provided to determine the effect of thermomechanical ice-coupling. The method we pursue (initially described in Pollard and DeConto, 2012) is to run the full thermomechanical coupling and adjust basal sliding coefficients during the run so as to obtain a steady-state ice sheet where the difference between modelled and observed ice geometry (thickness) is minimized. The result is a thermodynamic field that is in agreement with the ice dynamics, which has definite benefits when running the model over longer time periods. The method is often called a nudging method or a spinup with ice thickness targets (Seroussi et al., 2019; 2020). Generally speaking, our misfits are RMSE=50m for ice thickness and RMSE=100m/a for ice velocity. The advantage of having a thermomechanical coupling with this method is that the temperature field can adjust during the forward run, which is not the case with models that optimize ice stiffness and keep this constant over time. For the purpose of our study, where thermodynamic effects influence basal hydrology, this approach is preferred.**

Technical comments:

l. 55: maybe "the ability to resolve subglacial processes is limited by models' coarse spatial and temporal scales, thus large-scale representations of such processes (refs) are usually implemented in continent-scale ice sheet models", if it does not distort the intended sense. I got a little hung up on this sentence.
**Thanks for this. We have modified it as: "In addition, the ability to resolve subglacial processes is often limited by the coarse spatial and temporal scales used in models, hence large-scale representations of such processes are usually implemented in continental-scale ice sheet models (refs)."**

l. 46: mathematically effective pressure is not a boundary condition for the friction law, but rather a "state variable", meaning it defines the state of the drainage system that in turns affects basal friction through the friction law. Accordingly, I would rephrase as "The effective pressure is then used as a state variable in the common …"
**We have corrected this.**

l. 71: " an ice sheet that are applied …". Then, remove "in theory", and rephrase as "The effective pressure N is defined as …"
**We have corrected this.**

l. 86: terminated -> terminating
**Corrected.**

l. 95: " based on a single drainage element type", or at least, so I believe. Can't make sentence of the sentence as is
**Corrected.**

l. 105: lead to spontaneous water flow -> drive water flow
**Corrected.**

l. 122: need some justification for how $d^0_w$ is chosen, and whether/ how it affects model results
**This limit value has been determined based on the modelling of the subglacial water depth across the whole ice sheet. $d^0_w$ is then chosen as the maximum value of this field, similar to the approach in Bueler and Brown (2009).**

l. 248: increasing -> increased (? not sure)
**We have changed this into 'a higher sensitivity'.**

l. 317: "the power in the power law" - it is not immediately obvious, without knowing the literature, if you are talking about m or p. Please clarify.
**We have defined that this concerns $m$.**

—————————————————-

RC1
General comments

The paper is a well-written investigation into the the respective effects of exponents in the sliding law versus those of coupling with (simple) subglacial hydrological models on the evolution of the Antarctic Ice Sheet up until 2100. It tests four sliding-law exponents (m=1,2,3,5) and four different subglacial hydrology approaches, as well as a no-hydrology approach, across two extreme forcing scenarios and three realistic forcing scenarios. The authors conclude that, at the ice-sheet-wide scale, the exponent in the sliding law has a larger impact than the choice of subglacial hydrology model; this choice only modulates the eventual mass loss value up and down slightly. At a basin level, however, this finding is less evident.

Overall, I think the paper is well-written and logically structured, as well as being sound science. The figures are well-presented and clear. I do, however, have a couple of reservations that I would like to see addressed before recommending the paper for publication, which I detail below. Specifically, I question the values of some parameters, and the relationship between the presented results and conclusions.

**We thank the referee for the effort in reviewing our manuscript and for the positive comments. Below we answer the specific comments in more detail (in bold).**

Specific comments

I have two areas of specific concern with the paper that I feel may need some improvement before publication:

1. The choice of the maximum saturated till layer thickness parameter. This parameter is shown in the results section to be of considerable importance in the behaviour of the ice sheet in the TIL scenario (and this is discussed towards the end of the discussion section), yet the choice of 2 m as the value used is never justified or referenced. If it is a value that has been taken from the literature, please include appropriate references; if not, please justify why this value was chosen (sensitivity tests, model stability, …). I suspect that, ultimately, it will not make a huge difference to the conclusions of the paper, but I feel this aspect needs to be better explained.

   **This value is taken from Bueler and Brown (2009) and Bueler and Van Pelt (2015) and is used in the standard PISM model. In Bueler and Van Pelt (2015) the same limits have been applied to the Greenland ice sheet and the results (conform ours) are shown in their Figure 7. Areas at the pressure melting point have a W_til value of 2 m (saturated). Other values of W_til (max) would lead to a similar result, if the water fill up in the till is larger than the till drainage. Since subglacial conditions in Antarctica are relatively stable over the time periods we considered, there is not much we expect to change, like the Greenland experiment in Bueler and Van Pelt**

(2015). Nevertheless, we realize that this approximation is rather crude, but the overall idea of our paper was to test existing and relatively simple hydrological models to be applied to the Antarctic ice sheet. We are currently developing more exhaustive treatments. To comply with the remark, we have added  "This value is taken from \citet{bueler09} and \citet{VanPelt2012} as is used in the standard PISM model.".

2. The link between the presented results and conclusions. The authors conclude that the choice of the sliding law exponent is of much greater importance at the ice-sheet scale than the choice of the subglacial hydrology model in determining the evolution of the ice sheet. I agree that this is strongly supported by the results from the extreme-forcing scenarios (ABUK and ABUM), but this does not seem to be the case for the ISMIP6 experiments (compare, for example, Figure 5 with Figures 6-8), where the subglacial hydrology seems to be at least as important, if not more so, than the sliding law exponent (the range, for a given exponent, between the mass-loss values for the different subglacial hydrology models looks to be equal to or greater than the range for different exponent values for a given subglacial hydrology model). The lack of a clear relationship at the basin scale is noted and discussed, but not at the ice-sheet-wide scale. I feel therefore that the discussion could benefit from an additional paragraph addressing this contrast between the extreme- and realistic-forcing scenarios, along with a modification of the abstract and conclusion to acknowledge this.

We thank the referee for this observation. ABUMIP gives dominant mass loss, so the sensitivity to overall mass loss is easier to gauge. However, the ISMIP6 experiments lead to a variety of responses on the pan-Antarctic scale, for which it is not obvious to derive what makes the difference in response for the different basal hydrologies. These responses are not only reflecting differences in hydrology, but also other interactions with forcings, such as increased accumulation rate across vast areas of the East Antarctic ice sheet. This is the reason why we put the emphasis on the basin approach. However, we agree with the referee that more could be said on the difference between ABUMIP and ISMIP6 responses for different values of m in the manuscript, the abstract and conclusions.

In the manuscript we wrote in the discussion section: "A clear relationship between ice sheet sensitivity to forcing and the power in the basal sliding law is found for the ABUMIP experiments, leading to a higher sensitivity to climate forcing for more plastic sliding laws \citep{Sun2020}. Such a clear relationship is lacking in the ISMIP6 forcing experiments. This is probably due to a more complex interaction between the ice sheet and the forcing, as not only increased temperatures lead to more sub-shelf melt, but also to increased surface accumulation \citep{Seroussi2020}. The latter may therefore offset mass loss due to grounding line retreat in certain Antarctic drainage basins.".

**In the abstract we rephrased: "Here we show that spatial and temporal changes in water pressure or water flux at the base modulates basal sliding for a given power, especially for high-end scenarios, such as ABUMIP.", and added a final sentence: "This dependency is, however, less clear under realistic forcing scenarios (ISMIP6)."**

**We adapted the conclusions accordingly: "Spatial variation of effective pressure $N$ modulates basal sliding for a given value of the power-law exponent, where reduced effective pressure in the grounding zone generally increases the ice sheet sensitivity, especially for high-end scenarios, such as ABUMIP. This dependency is, however, less clear under realistic forcing scenarios (ISMIP6)."**

Technical comments

*We will correct these in the revised manuscript.*

- p.1,l.3: 'classic' for 'classical'

  **Corrected.**

- 1,l.5: 'the above' for 'above'
  **Corrected.**
- 1,l.5: remove 'i.e.' and replace with a colon
  **Corrected.**
- 1,l.8: the sensitivity of the ice sheet in what sense? You make it clear at the end of the sentence, but it might be worth adding 'to climatic forcing' or something similar here to clarify things a little.
  **We added 'to climatic forcing'.**
- 1,l.10 'modulate' not 'modulates'
  **Corrected.**
- 1,l.11: an increased sensitivity of what to what? Again, it's fairly obvious you mean of the ice sheet to climatic forcing, but it bears restating, especially here in the abstract, just so it's really clear.
  **We added 'to climatic forcing'.**
- 1,l.14: 'store' for 'storage'
  **Corrected.**
- 1,l.15: 'from', not 'of'
  **Corrected.**
- 1,l.23: 'laws'
  **Corrected.**
- 1,l.24: delete the second occurrence of 'conditions'
  **Corrected**

- 2,l.25: the sensitivity of ice-sheet flow to what?
  **Again we added to climatic forcing to make this clear.**
- 2,l.29-30: replace the 'either...or...' phrasing with a 'both...and…' one
  **Corrected.**
- 2,l.36: 'the spatial and temporal scale'
  **Corrected together with suggestions of the editor.**
- 2,l.40-41: replace 'and to gauge' with 'nor the gauging of'
  **Corrected.**
- Table 1: You refer to Cd here for the till drainage rate, but then talk about Ct later in Section 2.1.4. Pick one and make it consistent across the table and the text.
  **Corrected: kept with C_t**
- 8,l.164-165: Is the historical run sufficient as a relaxation run? I think it would be good to include a sentence here justifying why you don't need to do an actual relaxation run (or to change the phrasing, because it sounds very casual at the moment – something like 'and thus also serves as a relaxation run' would sound better)
  **Actually, the historical run can be considered a short relaxation run, but is for the ISMIP6 experiments forced by atmospheric and ocean forcing over this period of ten years for which climate was rather constant. During this period, ice sheet changes remain rather limited (order of model drift). We rephrased according to the suggestion of the referee.**
- 8,l.165: replace 'i.e' with a colon
  **Corrected.**
- 13,l.223-224: This description of Fig. 6 is a little confusing. The NON model run (i.e. no subglacial hydrology coupling) shows zero mass loss or a slight gain, but most of the model runs with some form of subglacial hydrology show near-zero or a slight mass loss, contrary to what the text says. I think you may have meant 'without' rather than 'with' in l.223, but, even then, the text would give the impression that mass gain is the rule, rather than the exception. I'd suggest re-wording this description slightly to make things clearer.
  **Indeed, it should be without. Thanks for spotting this. We also rephrased this.**
- 15,l.254: I'd replace 'is more uncertain in altering' with 'has a more uncertain effect on'. I would also add, at the end, 'than is seen at an ice-sheet level'
  **Corrected.**
- Figure 10 caption: 'Line colors' for 'Lines color'
  **Corrected.**
- Figure 10: what are the red lines on the two left-most panels (are they GL position, but, if so, why are there none on the two middle panels?)
  **The colored lines in the left and middle panels are the grounding positions corresponding to the different hydrological models (colors corresponding to the graphs on the right panels). However, since most of the grounding line positions coincide, mostly the black line is seen and the others are hidden behind. We mention this in the figure caption now.**

- Figure 10: make it clear that the Schoof runs are the runs presented in the results section earlier. I was a little confused and it took me a while to work out what was going on.

  **We mention in the caption: "solid lines represent a grounding-line flux parameterization due to \citet{schoof07a} (standard runs) and dashed lines due to \citet{tsai15}. Grounding-line positions in the left and middle panels are largely overlapping."**

—————————————

RC2

In the manuscript, the authors study the sensitivity of the Antarctic Ice Sheet model to the choice of subglacial hydrological model and to the values of power exponent in the Weertman/Budd sliding law. The authors conduct two series of numerical experiments, considering extreme and realistic environmental forcings in the ABUMIP and ISEMIP6 setups, respectively. One of the novel findings presented in the study is the increased sensitivity in case when the subglacial model depends on the subglacial water pressure.

The paper in question is definitely of scientific interest, is well-written, and I would recommend it for publication after minor revisions. I have two general comments, detailed below, followed by specific comments/questions.

**We thank the referee for the effort in reviewing our manuscript and for the positive comments. Below we answer the specific comments in more detail (in bold).**

General comments:

1. The subject of the study is the sensitivity of the sliding laws and various subglacial hydrological approaches. However, the sensitivity is not formally defined in the text. This makes it difficult to follow the discussion and to reason about the results of the paper. I therefore suggest the authors to define the sensitivity quantitatively and to use that definition throughout the text in a consistent way. An additional figure presenting the summary of the sensitivity study for the ice sheet scale would also simplify the interpretation of the results.

**Thank you for this pertinent remark. Indeed, we could have done a better job in properly defining what is meant by sensitivity. A higher sensitivity means an overall larger mass loss for the same given forcing. In order to clarify this (see RC1) we mention in all cases "sensitivity to climatic forcing". We also clarified this better in the results section for the ABUMIP experiments, where the second paragraph starts with: "Incorporating hydrology in the model generally leads to a higher sensitivity, where SWD and HAB show a higher mass loss due to the applied forcing (loss of ice shelves). "**

**However, we do not really understand what is meant by a summary figure for the sensitivities on the ice sheet scale, as the figures that are in the manuscript**

**already represent these. Putting it into one single figure is more problematic given the difference in scale.**

2. One of the factors that determine the dynamics of the ice sheet is the basal sliding coefficient $A_b$ first used in the Eq. 1. In the paper, the spatial distributions of $A_b$ are obtained through the optimization procedure for every combination of the power exponent m and the model for subglacial hydrology. Are these values of the basal sliding coefficient constrained in any way, e.g., in order to be within physically plausible ranges? How these values depend on the choice of m? I would recommend providing the figure(s) presenting the spatial distributions of $A_b$ at least for some representative problem setups and discussing how the values and spatial variation of $A_b$ influence the response of the ice sheet both on large scale and basin scale.

**For each value of m, the range of values of Ab is of course different, as the coefficients Ab are a multiplier to the sliding law. Nevertheless, we try to avoid overfitting and let the coefficient Ab evolve over maximum 4 to 5 orders of magnitude, leading to basal sliding velocities that are within the physical range (from mm/a to hundreds of m per year). The pattern of Ab is broadly consistent for different values of m and/or different subglacial hydrological approaches (their absolute value is however different of course). The highest values are encountered in outlet glaciers and ice streams and along the Siple Coast; the lowest values are within the interior of the East Antarctic ice sheet. The pattern is very similar to what is presented in Pollard and DeConto (2012), where the optimization method was presented initially. In order to prevent overfitting, we use a low-pass filter as regularization. This is also one of the reasons that the overall pattern is quite similar between the different experiments. We now added a Supplementary figure displaying the optimized distribution of Ab for the different hydrological models and for m=1 and m=3. We added a section in the discussion on this:**

**"For the different representations of the effective pressure $N$ in this paper, the pattern of the optimized slip coefficients $A_b$ is rather similar, albeit that their absolute values may differ, especially for the TIL model (Supplementary Figures S1 and S2). Overall, the highest values of $A_b$ are found in the Siple Coast, Amundsen Sea area, and the Recovery basin, where also the fastest ice flow occurs. Low values of effective pressure $N$ in these areas only partially alter the value of the optimized slip coefficients, but do not substantially change the pattern of $A_b$. The same pattern is also obtained for different values of the power in the power law $m$, demonstrating that the optimization scheme is rather robust. The use of a low-pass filter for the regularization to avoid overfitting also helps the robustness of the method (refs.)."**

Minor comments/questions:

*Below we answer those questions that need some explanation. In the revised manuscript we will take care of the corrections and typos that are asked.*

- 1, l. 7 - please define "RCP" before first use;
  **Corrected.**
- 6, l. 118 - how $Q_l$ is calculated?;
  **$Q_l$ is calculated as the incoming flux plus the basal melting rate corrected for the unit width of the cell or the subgacial water speed mutliplied by the subglacial water thickness. For more details on the method, see for instance LeBrocq et al. (2006) where the same method is used for determining balance fluxes of ice. We have added this in the text.**
- 6, l. 125 - "and the subglacial water flux, i.e.," - change to "and the subglacial water flux $\varphi$, i.e.,";
  **Corrected.**
- 6, Eq. 10 - please define $A_o$, e.g., "and $A_o$ the initial value of $A_b$, obtained through a nudging method described in Section 3";
  **Corrected.**
- 6, l. 131 - "the effective pressure N is considered constant for SWF" - what is the value of the effective pressure N? Does this value influence the results?;
  **In the model it is taken constant (used as a scaling factor). However, for the manuscript it is better to remove this statement and just state that for SWF, the effective pressure is not considered in the sliding law.**
- 7, l. 137 - please define the "yield stress" of what is discussed;
  **It is the yield stress defined in the equation (1). We modified the sentence by "A fixed fraction of ice overburden equal to one implies an effective pressure and consequently a yield stress equal to zero (Equation (1))".**
- 7, l. 145 - W instead of $W_{til}$;
  **Corrected.**
- 7, l. 153 - "$\delta p_o$ is the lower bound on N, taken as a fraction of the ice overburden pressure." - I suggest changing it to "$\delta p_o$ is the lower bound on N, taken as a fraction $\delta$ of the ice overburden pressure $p_o$" for better readability;
  **Corrected.**
- 15, l. 258 - "between difference" should read "between different";
  **Corrected.**
- 16, Fig. 9. - the TIL model seems to be dramatically different from other models for m > 1, especially for the Enderby Land basin (Fig. 9c). It would be useful to see an explanation for this.
  **It should be noted that the y-axis has a different scale than in Pine Island and Thwaites and in Wilkes basins. The main reason why the TIL model is less sensitive in Enderby Land is because of the lack of saturated till in this basin (fig. 4). Furthermore, the TIL model is generally less sensitive than the other approaches. We have added this in the text.**

---

## Referee Report (RR1)

**Second Review of Kazmierczak et al. (2022) 'Subglacial hydrology modulates basal sliding response of the Antarctic ice sheet to climate forcing'**

**General comments**

The paper remains a well-written investigation into the the respective effects of exponents in the sliding law versus those of coupling with (simple) subglacial hydrological models on the evolution of the Antarctic Ice Sheet up until 2100. It tests four sliding-law exponents (m=1,2,3,5) and four different subglacial hydrology approaches, as well as a no-hydrology approach, across two extreme forcing scenarios and three realistic forcing scenarios. The authors conclude that, at the ice-sheet-wide scale, the exponent in the sliding law has a larger impact than the choice of subglacial hydrology model; this choice only modulates the eventual mass loss value up and down slightly. At a basin level, however, this finding is less evident. Overall, I think the paper is well-written and logically structured, as well as being sound science. The figures are well-presented and clear.

The authors have clearly put additional work into the paper to meet the points I raised in my initial review, for which I am grateful. In particular, I feel the conclusions now fit better with the results presented in the paper, my previous chief concern. As a result, I see no reason to further delay publication of the paper and recommend that it be accepted.